# Health effects of immediate telework introduction during the COVID-19 era in Japan: A cross-sectional study

**Qian Niu**[1], **Tomohisa Nagata**[2], **Naoto Fukutani**[1,3], **Masato Tezuka**[3,4], **Kanako Shimoura**[1], **Momoko Nagai-Tanima**[1], **Tomoki Aoyama**[1]*

**1** Department of Physical Therapy, Human Health Sciences, Graduate School of Medicine, Kyoto University, Kyoto, Japan, **2** Department of Occupational Health Practice and Management, Institute of Industrial Ecological Sciences, University of Occupational and Environmental Health, Japan, Kitakyushu, Japan, **3** BackTech Inc., Kyoto, Japan, **4** Department of Public Health, Graduate School of Health Science, Kobe University, Kobe, Japan

* blue@hs.med.kyoto-u.ac.jp

## Abstract

### Background

Telework has been widely discussed in several fields; however, there is a lack of research on the health aspects of teleworking. The current study was conducted to determine the health effects of teleworking during an emergency statement as evidence for future policy development.

### Method

This was a cross-sectional study in which we administered an online questionnaire to 5,214 general workers (response rate = 36.4%) from June 2020 to August 2020. Based on working methods during the pandemic, workers were categorized into the office group (n = 86) and telework group (n = 1597), and we characterized their demographics, changes in lifestyle, telework status, physical symptoms, and mental health.

### Results

The results showed that the workers' residence, marital status, management positions, and employee status affected the choice of the work method. During the emergency, teleworkers experienced more changes in their habits than office workers. In terms of exercise habits, 67.0% of the individuals belonging to the office-telework (OT) group exercised less. Approximately half of the teleworkers were satisfied with their telework, and those in the OT group were less satisfied with their telework than those in the telework-telework (TT) group, and they reported an increase in both working hours and meeting hours. Work-family conflict was more pronounced in the TT group than in the two other groups. Only 13.2% of individuals did not experience any stress in the past 30 days, and all three groups showed varying degrees of anxiety and depressive tendencies. In addition, all teleworkers experienced adverse physical symptoms before and after the emergency.

**Data Availability Statement:** We submitted an application to the Ethics Committee, stating that the researcher would be responsible for storing the data. As a result, our research has been approved

by the Ethics Committee. The ethics committee has not approved the release of the data to the public. For the reasons mentioned above, we are unable to make the data widely available. We have provided the contact information of the ethics committee for data inquires: Ethics committee contact information as follows: Department of Occupational Health Practice and Management Institute of Industrial Ecological Sciences University of Occupational and Environmental Health, Japan E-mail: j-skeiei@mbox.med.uoeh-u.ac.jp.

**Funding:** The authors received no specific funding for this work.

**Competing interests:** The authors have declared that no competing interests exist.

**Abbreviations:** OO, office-office, people who worked in offices before and after the emergency statement; OT, office-telework, people who switched from office to telework because of the emergency statement; TT, telework-telework, people who continued teleworking unrelated to COVID-19; MSDs, musculoskeletal disorders.

## Conclusion

Health issues associated with teleworking should be given adequate attention.

## Introduction

### Background

Telework is gaining importance with the evolution of communication technologies. The complex and multifaceted literature on telework spans multiple study fields and has implications for stakeholders [1]. Telework has various positive and negative effects, which can lead to opposed conclusions from different studies, even based on the same topic. Moreover, in the background of the global pandemic, telework is attracting attention as an important alternative to traditional office work.

### Advantages of telework

Studies have shown that full-day telework results in fewer daily travels and avoids rush-hour traffic congestion [2, 3]. Henke et al. investigated the impact of telework intensity on employee health and concluded that employees can benefit from telework opportunities, suggesting that the health risks of teleworking are related to work intensity. Non-teleworkers had a higher risk of health indicators (e.g., obesity, alcoholism, physical inactivity, and smoking), and teleworkers were less likely to be depressed [4]. A review of the literature related to ergonomics and telework by de Macedo [5]showed that teleworking is an effective tool for work-life balance and contributes to well-being. Gajendran [6] conducted a meta-analysis of 12,883 employees by constructing a theoretical framework and concluded that teleworking offers greater autonomy and lowers conflict between work and family. It also showed favorable effects on job satisfaction, performance, replacement awareness, and role stress. In addition, intensive telework (more than 2.5 days per week) was also beneficial in lowering work-family conflict, and Jostell [7] also found that after-hours telework was effective in completing unfinished work and did not create work-family conflict. Hornung [8] et al. provided further evidence, based on previous research, to support the idea that telework improves an employee's quality of life. Restrepo et al. [9] concluded that teleworkers may have more time to prepare and consume food at home, and as home-cooked food has lower calories and higher nutrition, it may provide health benefits for teleworkers. Several studies have shown that teleworking can provide job opportunities for vulnerable people such as postpartum women [10], the elderly [11], and people with disabilities [12–15].

In short, teleworking helps avoid traffic congestion, increases work independence and satisfaction, reduces health risks and depressive tendencies, maintains work-life balance, improves the quality of life and well-being, and facilitates re-employment of vulnerable people thereby achieving social equality.

### Disadvantages of telework

Studies by Kitou et al. [16] and O'Brien et al. [17] have shown that teleworking offsets the beneficial effects of reduced commuting by working longer hours at home, increasing energy use, and transportation. Zhang et al. [18] analyzed telework from the perspective of work-life conflict and showed that the association between life dimensions (sex, marital status, and parenthood) and telework behavior is complex. Children play a crucial role in telework performance,

which not only increases work-to-family conflict and family-to-work conflict but also triggers the redistribution of household chores within couples and exacerbates gender differences. Telework is flexible as an alternative and complement to traditional office work; however, studies by Steidelmuller and Ahmed [19, 20] revealed a correlation between telework and consistent attendance during illness, further compromising the health of teleworkers. Golden et al. [21] stated that occupational segregation induced poor job performance and reduced the intention to move as telework accumulated. Glass et al. [22] also argued that telework, which occurs mostly in the low-yield overtime component, did not increase the real flexibility of employees to work when and where they want but rather largely lengthened the workday and encroached on home and family time.

In summary, teleworking, under certain conditions, has negative effects in areas such as energy use, occupational isolation, and work-family balance, which affect the implementation of teleworking and the well-being of teleworkers.

## Telework during the pandemic

As the number of infections in Tokyo increased dramatically, a state of emergency was imposed in Japan [23, 24] from April 7, 2020 to May 25, 2020 to prevent the spread of the severe acute respiratory syndrome coronavirus 2. The new normal was introduced on May 26 with the maintenance of social distancing, and the number of infections subsided; however, a renewed increase in human activity during the economic recovery may increase the number of infections. Karako et al. [25] adopted a localized stochastic transition model. The results of an over-model analysis for Tokyo showed that for Tokyo and other densely populated cities, in the face of the new normal of the coronavirus disease 2019 (COVID-19), shifting the work patterns of 55% of office workers aged 20–64 years to telework and reducing the time spent in high-risk areas by 25% could control the spread of infection without causing significant economic losses. Fisher et al. [26] conducted a case-control survey to compare telework between polymerase chain reaction-positive and negative individuals, and found that the results showed that the former were more likely than the latter to attend work in a public office 2 weeks before the onset of infection. Consistent with Karako's [25] conclusion, teleworking reduces the risk of infection.

Although telework is employed as an important solution to outbreaks, we consider that isolation may have negative effects on individual and public health. The ability to make sound policy recommendations regarding telework depends on abundant and qualitative study results. Surprisingly, little research has been conducted on the association between telework and health-related performance [1], especially during the pandemic. This study aimed to determine the health problems associated with teleworking during a pandemic, thereby improving the well-being of teleworkers and providing evidence to support subsequent related studies.

## Methods/design

### Study design

This was a cross-sectional study using an online survey conducted after the emergency statement due to COVID-19.

### Study participants

All the participants in this study were general workers from companies that had business contact with BackTech Inc. The survey was administered to 5,000 subjects, aged 20–80 years, regardless of the sex.

## Procedure

This study was jointly conducted by the University of Occupational and Environmental Health, Japan (UOEH-U), BackTech Inc., and Kyoto University (Kyoto-U). BackTech Inc. designed the questionnaire with UOEH-U, recruited the participants, and collected and shared the anonymized data. After that, Kyoto-U analyzed the data according to the study purpose and shared the results of the analysis with the other institutions. Eventually, Kyoto-U completed the writing and contributions of the paper.

This study was approved by the Ethics Committee of UOEH-U. Data were collected using an online convenient questionnaire tool, Survey Monkey. The survey was conducted from June 7, 2020 to September 7, 2020, a period when we entered the "new normal" after the emergency statement. The survey tool automatically verified that all questions had to be answered before submission and that they could not be submitted twice. To ensure the validity and accuracy of the collected data, all the subjects were prior-informed about the study, and signed, written informed consent was obtained online before answering the questions.

## Measures

The following six dimensions were investigated after the emergency statement: demographics, lifestyle changes, telework characteristics, mental health, physical symptoms, and work-family conflict. All variables are displayed in descriptive statistics.

**Demographics and lifestyle changes.**   Participants were asked to provide basic demographic information, including age, sex (male/female), residence (Tokyo Metropolis/other), marital status (married/other), cohabitant status (yes/no), managerial position (yes/no), and employment status (regular staff/other). According to the health examination items of the Ministry of Health, Labor, and Welfare, Japan, we investigated the participants' smoking habits (yes/no), exercise habits before and after the emergency statement (yes/no), sleeping hours (hours), drinking habits (yes/no), and their changes (decrease/increase/no change).

**Telework.**   For those who switched from office to telework due to the emergency statement, we investigated the characteristics of teleworking. For teleworking, we surveyed the number of telework days per week before and after the emergency statement (0/1-2/3-4/5), changes in work hours and meeting hours (decrease/increase/no change), telework satisfaction (satisfied/neither/dissatisfied), working environment (dedicated workspace/other), the types of seat (chairs/other), and inconvenient items.

**Work-family conflict.**   Work-family conflict is a source of stress experienced by many individuals. Carlson et al. [27] constructed and validated a multidimensional scale (Work-Family Conflict Scale [WFCS]) to measure work-family conflict. Watai et al. [28] examined the reliability and validity of the Japanese version of the WFCS. To lower assessment time and participant fatigue, and to increase the response rate, Russell and Lisa [29] developed an abbreviated version in which two 3-item abbreviated measures based on Carlson's multidimensional measures, one to assess work-to-family conflict, and one to assess family-to-work conflict. We used the short version (translation in Japanese) to investigate the work-family conflict.

**Depression and anxiety.**   The 6-item Kessler Psychological Distress Scale (K6) has been used in recent years in international multicenter large-sample community population epidemiological studies to screen for mental disorders, mainly anxiety and depression. Currently, it is the simplest and most efficient screening tool [30, 31]. Participant mental health was surveyed using the Japanese version of the K6 developed by Furukawa et al. [32]. In addition, we surveyed participants about their most stressful items in the past 30 days.

**Physical symptoms.**   We investigated the following 15 physical symptoms: stiff neck, eye strain, back pain, fatigue, feeling of heaviness in the body, headache, diarrhea, constipation,

dizziness, tinnitus, frequent urination, cough and sputum, joint pain, hearing loss, and numb fingers and forearms. Physical symptoms were analyzed using a cumulative score, which was 1 if the symptom was present and 0 if it was not, with each participant's total score ranging from 0 to 15.

### Data processing and analysis

Data were analyzed using the SPSS software (version 26.0). The Mann-Whitney U test was used to compare the ordered categorical variables before and after the emergency declaration. Therefore, we compared the group of those who insisted on working in the office (office-office [OO]), the group of those who transformed from office-work to telework (office-telework [OT]), and those who continued working from home independent of the outbreak (telework-telework [TT]). A p-value of <0.05 was considered statistically significant.

## Results

### Demographics and lifestyle changes

A total of 5,214 participants were invited, and 1,896 individuals answered the questionnaire (response rate of 36.4%). As shown in Table 1, the number of people in the OO, OT, and TT groups were 86, 1597, and 213, respectively. There were no statistical differences in the age and sex among the three groups, with an average age of 44.52 (11.2) years for all, and 76.8% of the employees were males. The remaining demographic indicators were statistically different, such as residence, marital status, cohabitant presence, managerial position, and employment status. In the OT group, 30.2% of the participants lived in the Tokyo metropolitan area, 66.8% were married, 70.1% reported having cohabitants, 40% were in managerial positions, and 88.7% were regular staff, all of which were greater than those of the OO group. However, the TT group had the highest percentage in terms of being married, presence of cohabitants, and being a regular staff among the three groups (77.5%, 79.8%, and 96.2%, respectively).

Regarding lifestyle habits, there were no statistical differences in smoking habits, exercise habits, and drinking habits in the OO, OT, and TT groups before Japan the emergency statement was declared, while the drinking habits were statistically different, with p-values <0.01. After the emergency statement, there were no changes in overall smoking habits, exercise habits, sleep duration, alcohol consumption frequency, and alcohol consumption (55.8%, 21.6%, 64%, 60.5%, and 61.3%, respectively). The number of smokers in the OT group increased by 28%, the number of exercisers decreased by 67.3%, the sleep duration increased by 30%, and the frequency and amount of alcohol consumption reported no change by 59.6% and 63.6%, and a decrease by 26.7% and 25.5%.

### Characteristics of teleworkers during the COVID-19 era

Of the 1,896 people who responded to our survey on the status of telework, 1,597 (84.2%) had switched from office to telework, and 213 (11.2%) teleworked unrelated to COVID-19 (Table 2). Approximately 50% of teleworkers reported an increase in work hours and meeting hours. Overall, teleworkers were satisfied with their telework status. Only 37.9% of the individuals belonging to the OT group had a professional office environment, lower than that of the TT group (45.1%), and the rest chose a dining table or another simple environment. In both groups, 74.9% and 80.3% of the individuals sat on chairs to work. Faced with the sudden outbreak, only 18.8% of the OT group reported no inconvenience, which was lower than that of the TT group (37.6%). The top four inconveniences were the unprepared desktop environment, no information tools, no workroom, and slower Internet speeds.

**Table 1. Recruitment and demographic, lifestyle changes during emergency statement by telework or not[1].**

| Characteristics | All | Office-Office | Office-Telework | Telework-Telework | p |
|---|---|---|---|---|---|
| **n** | 1896 | 86 | 1597 | 213 | |
| **Demographic characteristics** | | | | | |
| Age(yo) (mean (SD)) | 44.52(11.2) | 45.44(12.9) | 44.10(11.3) | 46(8.9) | 0.456 |
| Gender(male)(n(%)) | 1456(76.8) | 63(73.3) | 1226(76.8) | 167(78.4) | 0.633 |
| Residence(Tokyo)(n (%)) | 528(27.8) | 2(2.3) | 483(30.2) | 43(20.2) | <0.01** |
| Married(yes) (n(%)) | 1274(67.2) | 43(50) | 1066(66.8) | 165(77.5) | <0.01** |
| Cohabitants(yes)(n(%)) | 1343(70.8) | 54(62.8) | 1119(70.1) | 170(79.8) | <0.01** |
| Manager(yes) (n(%)) | 710(37.4) | 12(14) | 639(40) | 59(27.7) | <0.01** |
| Employment status(regular staff)(n (%)) | 1682(88.7) | 60(69.8) | 1417(88.7) | 205(96.2) | <0.01** |
| **Lifestyle characteristics before the emergency statement** | | | | | |
| Smoking status (yes)(n (%)) | 355(18.7) | 19(22.1) | 304(19) | 32(15) | 0.505 |
| Changes during emergency statement(n (%)) | | | | | |
| Decrease | 69 (19.4) | 0 | 55(18.1) | 14(43.8) | / |
| Increase | 88(24.8) | 0 | 85(28) | 3(9.4) | / |
| No change | 198(55.8) | 19(100) | 164(53.9) | 15(46.8) | / |
| Exercise habits(yes)(n (%)) | 1110(58.5) | 42(48.8) | 924(57.9) | 144(67.6) | 0.363 |
| Changes during emergency statement(n (%)) | | | | | |
| Decrease | 736(38.9) | 4(9.5) | 622(67.3) | 110(51.6) | / |
| Increase | 134(7.1) | 14(33.3) | 113(12.2) | 7(3.3) | / |
| No change | 240(21.6) | 24(57.1) | 189(20.5) | 27(12.7) | / |
| Sleeping hours (mean (SD)) (hours) | 6.26(0.98) | 6.53(1.11) | 6.13(0.96) | 6.10(0.91) | 0.181 |
| Changes during emergency statement(n (%)) | | | | | |
| Decrease | 122(6.4) | 6(7) | 99(6.2) | 17(8) | / |
| Increase | 561(29.7) | 6(7) | 478(30.0) | 77(36.2) | / |
| No change | 1213(64) | 74(86) | 1020(63.9) | 119(55.9) | / |
| Drinking (yes)(n (%)) | 1237(65.2) | 45(52.3) | 1055(66.1) | 137(64.3) | <0.01** |
| Frequency changes during emergency statement(n (%)) | | | | | |
| Decrese | 492(25.9) | 27(31.4) | 426(26.7) | 39(18.3) | / |
| Increse | 257(13.6) | 1(1.2) | 219(13.7) | 37(17.4) | / |
| No change | 1147(60.5) | 58(67.4) | 952(59.6) | 137(64.3) | / |
| Alcohol consumption during emergency statement(n (%)) | | | | | |
| Decrese | 528(27.8) | 23(26.7) | 408(25.5) | 97(45.5) | / |
| Increase | 206(10.9) | 2(2.3) | 174(10.9) | 30(14.1) | / |
| No change | 1162(61.3) | 61(70.9) | 1015(63.6) | 86(40.4) | / |

[1] The sample size may vary for some variables, because of missing values. Means and standard deviations (SD) are presented for normally distributed continuous variables, number of people and percentages are presented for unordered categorical variables.

[2] Statistical differences among the three groups are derived from *Kruskal-Wallis H test*.

## Work-family conflict related to telework

As shown in Table 3, the unidirectional conflict between work-to-family and family-to-work did not differ statistically; however, the conflict between work and family was statistically significant with a p-value <0.01. In addition, the results of multiple comparisons between the three groups showed that only the result of the TT group was statistically significant, with a p-value <0.01. The results of the OO and OT groups were not statistically significant.

**Table 2. Characteristics of teleworkers during the COVID-19 era.**

| | Office-Telework | Telework-Telework |
|---|---|---|
| *n* | 1597 | 213 |
| **Telework days during the state of emergency (n (%))(days/week)** | | |
| 0 /week | 101(6.3) | 7(3.3) |
| 1–2 | 358(22.4) | 26(12.2) |
| 3–4 | 561(35.1) | 74(34.7) |
| 5 or more | 577(36.1) | 106(49.8) |
| **Changes of work hours(n (%))** | | |
| Decrease | 350(21.9) | 38(17.8) |
| Increase | 941(58.9) | 119(55.9) |
| No change | 306(19.2) | 56(26.3) |
| **Changes of meeting hours(n (%))** | | |
| Decrease | 270(16.9) | 29(13.6) |
| Increase | 809(50.7) | 105(49.3) |
| No change | 518(32.4) | 79(37.1) |
| **Telework satisfaction(n (%))** | | |
| Very satisfied | 216(13.5) | 51(23.9) |
| Satisfied | 594(37.2) | 102(47.9) |
| Neither | 536(33.6) | 54(25.4) |
| Dissatisfied | 208(13) | 6(2.8) |
| Very dissatisfied | 41(2.6) | 0 |
| **Telework work environment(n (%))** | | |
| Dedicated workspace | 605(37.9) | 96(45.1) |
| Dining table | 486(30.4) | 57(26.8) |
| A simple place (such as a sitting table) | 422(26.4) | 47(22.1) |
| Others | 84(5.3) | 13(6.1) |
| **Typles of seats (n (%))** | | |
| Sitting on a chair(excluding zaisu) | 1196(74.9) | 171(80.3) |
| Sitting on the floor(including zaisu) | 374(23.4) | 42(19.7) |
| Standing-up | 9(0.6) | 0 |
| Others | 18(1.1) | 0 |
| **Inconvenient items (n (%))** | | |
| Not ready of desk environment | 645(40.4) | 66(31) |
| No information equipment | 594(37.2) | 62(29.1) |
| No workroom | 521(32.6) | 47(22.1) |
| Slow Internet speed | 438(27.4) | 49(23) |
| Noise | 287(18) | 40(18.8) |
| Small workspace | 276(17.3) | 22(10.3) |
| Dim workspace | 95(5.9) | 5(2.3) |
| No Communication tools | 50(3.1) | 7(3.3) |
| Others | 186(11.6) | 24(11.3) |
| No inconvenience | 300(18.8) | 80(37.6) |

## Mental health and physical symptoms

As shown in Table 4, the results of the K6 scale after multiple comparisons showed that there was no statistical difference between the OO group compared with both the OT and TT groups, while the results of the OT and TT groups were statistically significant (p = 0.007).

**Table 3. Work–family conflict related to telework [1].**

| | All | Office-Office | Office-Telework | Telework-Telework | p |
|---|---|---|---|---|---|
| *n* | 1327 | 52 | 1108 | 167 | |
| **Work-to-family score (mean (SD))** | 8.1(2.6) | 8.1(2.1) | 8.1(2.6) | 8.1(2.7) | 0.935 |
| **Family-to-work score (mean (SD))** | 6.8(2.2) | 7(1.9) | 6.8(2.1) | 7.2(2.4) | 0.194 |
| **Work and family conflict score(mean (SD))** | 14.9(4.2) | 15.1(3.2) | 14.8(4.2) | 15.3(4.6) | <0.01** |
| **Pairwise comparisons of group** | | | | | |
| **OO-OT** | | | | | p = 1.000 |
| **OO-TT** | | | | | <0.01** |
| **OT-TT** | | | | | <0.01** |

[1] Means and standard deviations (SD) are presented for all the variables in this table.

Because of single, divorce or death, the response rate for this section is only **70.0%** of total people.

[2] *Kruskal-Wallis H test* are used to compare the difference among these three groups.

Overall, 64.2% of the survey participants reported no anxiety or depression, while 24.1%, 6.6%, and 5.1% experienced varying degrees of anxiety and depression, respectively. In the past 30 days, only 13.2% of the participants indicated that they had never been stressed, and the most important reasons for stress included the inability to go out for recreation, concern about COVID-19, lack of communication with colleagues, and work-life imbalance.

As shown in Table 5, the number of adverse physical symptoms increased in all three groups before and after the emergency and was statistically significant in the two teleworking groups (p<0.01). The most reported symptoms were frozen shoulders, eye strain, back pain, fatigue, a feeling of heaviness, headache, dizziness, etc.

## Discussion

Of the 1,896 people who responded to the questionnaire, 84.2% switched to teleworking because of the emergency statement. Our results show that residence, marital status, job position, and employment status all influenced the choice of the work method. In contrast to Zhang et al. [18], our study shows that married people are more likely to work remotely and that sex has no effect on the choice of telework.

Our results are inconsistent with the model predicted by Brand et al. [33]. They suggested that people tended to increase or maintain their exercise frequency during the pandemic; however, they ignored the effect of occupation on exercise preference. Experts believe that physical activity can eliminate the negative effects of isolation and strengthen the immune system. During the current pandemic, moderate-intensity and moderate amount of exercise should be performed [34]. Increasing health awareness can help people establish healthy life goals and promote indoor exercise [35]. Shariat et al. and Schwendinger et al. [36, 37] developed an evidence-based indoor exercise program for teleworkers to counteract musculoskeletal problems, anxiety, and depression caused by inactivity.

The percentage of teleworkers with drinking habits was 13.8% higher than that of office-based workers. Approximately 40% of those with drinking habits reported a decrease in the frequency of drinking and amount of alcohol consumed. We partially agree with the results of Jackson et al. [38], who indicated that smokers and drinkers were more likely to report attempting to quit smoking or reduce alcohol consumption after the COVID-19 lockdown than before; however, alcohol consumption increased after the lockdown. Oksanen et al. and Wardell et al. [39, 40], for example, examined motivations for drinking during the lockdown

**Table 4. Mental health after the emergency statement with K6 scale[1].**

| | All | Office-Office | Office-Telework | Telework-Telework |
|---|---|---|---|---|
| **n** | 1896 | 86 | 1597 | 213 |
| **p (Kruskal-Wallis H test)** | | 0.006** | | |
| **Pairwise comparisons of group** | | | | |
| OO-OT | | 0.698 | | |
| OO-TT | | 1.000 | | |
| OT-TT | | 0.007** | | |
| **K6 scores(n (%))** | | | | |
| 0–5: No depression / anxiety | 1218(64.2) | 59(68.6) | 1011(63.3) | 148(69.5) |
| 6–10: Maybe depression / anxiety | 456(24.1) | 19(22.1) | 397(24.9) | 40(18.8) |
| 11–13: Suspected of depression / anxiety disorder | 125(6.6) | 3(3.5) | 107(6.7) | 15(7) |
| ≥14: Suspected of severe depression / anxiety disorder | 97(5.1) | 5(5.8) | 82(5.1) | 10(4.7) |
| **The most stressful item in the past 30 days(n (%)):** | | | | |
| No pressure | 250(13.2) | 17(19.8) | 203(12.7) | 30(14.1) |
| Long working hours | 59(3.1) | 1(1.2) | 51(3.2) | 7(3.3) |
| Uneasy about employment | 69(3.6) | 4(4.7) | 61(3.8) | 4(1.9) |
| Economic pressure | 68(3.6) | 8(9.3) | 57(3.6) | 3(1.4) |
| Drastic changes in telework | 119(6.3) | 0(0) | 112(7) | 7(3.3) |
| Work-life balance | 188(9.9) | 2(2.3) | 163(10.2) | 23(10.8) |
| Lack communication with colleagues | 199(10.5) | 4(4.7) | 176(11) | 19(8.9) |
| Lack communication with cohabitants | 32(1.7) | 3(3.5) | 22(1.4) | 7(3.3) |
| Child support | 27(1.4) | 1(1.2) | 15(0.9) | 11(5.2) |
| Housework | 18(0.9) | 1(1.2) | 10(0.6) | 7(3.3) |
| Personal time reduction | 24(1.3) | 1(1.2) | 18(1.1) | 5(2.3) |
| Personal time increase | 55(2.9) | 1(1.2) | 51(3.2) | 3(1.4) |
| Can't go out for entertainment | 379(20) | 18(20.9) | 319(20) | 42(19.7) |
| Concerns about health due to COVID-19 | 209(11) | 17(19.8) | 166(10.4) | 26(12.2) |
| Concerns about health exclude COVID-19 | 72(3.8) | 2(2.3) | 65(4.1) | 5(2.3) |
| Do not know | 16(0.8) | 0(0) | 14(0.9) | 2(0.9) |
| Others | 112(5.9) | 6(7) | 94(5.9) | 12(5.6) |

[1] K6 is used to screen for mental disorders such as anxiety and depression, and is considered the most convenient and efficient screening tool. According to the frequency of symptoms, there are 5 levels from "no time" = 0 points to "all time" = 4 points. Total score is 0–24. Percentages are presented for all the categorical variables in this table.

[2] *Kruskal-Wallis H test* are used to compare the difference among the OO group, OT group and TT group.

and concluded that cyberbullying victimization and psychological distress during telework were major risk factors. Child-rearing obligations, depressed mood, social and physical isolation, and loss of income all contributed indirectly to drinking problems.

The majority (84.2%) shifted from traditional office work to teleworking from home during the emergency statement, and 61.2% did not work fewer days than they did before the emergency statement. Although previous studies have shown the flexibility and efficiency of teleworking, workers were not prepared for sudden changes in the work style. Space limitations and the difficulty of separating private and work time added to the challenge of teleworking. However, most workers (84.4%) were satisfied or indifferent to teleworking. De Croon et al. [41] also provided supporting evidence to our results that a relatively enclosed workplace can increase privacy and job satisfaction as well as improve productivity.

**Table 5. Physical symptoms before and after the emergency declaration[1].**

| | All | | Office-Office | | Office-Teleworkp | | Telework-Telework | |
|---|---|---|---|---|---|---|---|---|
| **n** | 1896 | | 86 | | 1597 | | 213 | |
| | Before | After | Before | After | Before | After | Before | After |
| **Physical symptoms index** | 1.25 | 1.65 | 1.06 | 1.23 | 1.25 | 1.68 | 1.31 | 1.63 |
| **p** | <0.01** | | 0.076 | | <0.01** | | <0.01** | |
| **Physical symptoms (n (%))** | | | | | | | | |
| Asymptomatic | 845(44.6) | 635(33.5) | 47(54.7) | 42(48.8) | 717(44.9) | 524(32.8) | 86(40.4) | 69(32.4) |
| Symptomatic | 1046(55.2) | 1261(66.5) | 39(45.3) | 44(51.2) | 880(55.1) | 1073(67.2) | 127(59.6) | 144(67.6) |
| Stiff neck | 566(29.9) | 644(34) | 17(19.8) | 15(17.4) | 474(29.7) | 550(34.4) | 474(29.7) | 550(34.4) |
| Eyestrain | 467(24.6) | 664(35) | 18(20.9) | 21(24.4) | 392(24.5) | 574(35.9) | 392(24.5) | 574(35.9) |
| Back pain | 323(17) | 485(25.6) | 11(12.8) | 13(15.1) | 280(17.5) | 417(26.1) | 280(17.5) | 417(26.1) |
| Fatigue | 294(15.5) | 400(21.1) | 13(15.1) | 14(16.3) | 254(15.9) | 347(21.7) | 254(15.9) | 347(21.7) |
| Feel heavy | 190(10) | 288(15.2) | 5(5.8) | 9(10.5) | 161(10.1) | 249(15.6) | 161(10.1) | 249(15.6) |
| Cough and sputum | 37(2) | 30(1.6) | 2(2.3) | 2(2.3) | 31(1.9) | 25(1.6) | 31(1.9) | 25(1.6) |
| Frequent urination | 46(2.4) | 52(2.7) | 2(2.3) | 3(3.5) | 41(2.6) | 44(2.8) | 41(2.6) | 44(2.8) |
| Constipation | 68(3.6) | 82(4.3) | 1(1.2) | 3(3.5) | 58(3.6) | 67(4.2) | 58(3.6) | 67(4.2) |
| Joint pain | 30(1.6) | 65(3.4) | 1(1.2) | 1(1.2) | 23(1.4) | 56(3.5) | 23(1.4) | 56(3.5) |
| Headache | 102(5.4) | 116(6.1) | 6(7) | 7(8.1) | 82(5.1) | 96(6) | 82(5.1) | 96(6) |
| Diarrhea | 83(4.4) | 95(5) | 5(5.8) | 5(5.8) | 70(4.4) | 81(5.1) | 70(4.4) | 81(5.1) |
| Dizziness | 56(3) | 76(4) | 4(4.7) | 7(8.1) | 44(2.8) | 60(3.8) | 44(2.8) | 60(3.8) |
| Hearing loss | 27(1.4) | 28(1.5) | 2(2.3) | 2(2.3) | 23(1.4) | 24(1.5) | 23(1.4) | 24(1.5) |
| Numb fingers and forearms | 18(0.9) | 31(1.6) | 1(1.2) | 0(0) | 13(0.8) | 26(1.6) | 13(0.8) | 26(1.6) |
| Tinnitus | 56(3) | 69(3.6) | 3(3.5) | 4(4.7) | 48(3) | 56(3.5) | 48(3) | 56(3.5) |

[1] Percentages were presented for all the unordered categorical variables in this table.

[2] The physical symptoms are analyzed by cumulative scores, present = 1, absent = 0, Minimum score is 0, maximum score is 15; ***Kruskal-Wallis H test*** is used to compare the physical symptoms before and after the emergency statement; physical symptoms index = total score/n.

Statistical results did not show a difference in the mental health between workers who continued working in the office during the emergency period and those who switched to telework. However, people who teleworked for a long period showed more severe anxiety and depression compared with those who worked for a short period. Approximately 40% of teleworkers showed depressive and anxiety tendencies. Those who switched to teleworking were more likely to report distress resulting from work-family conflict, being in confined spaces at home, and worries about the COVID-19 pandemic. Many teleworkers have been unexpectedly asked to work remotely from home due to the pandemic, blurring the lines between their work and personal lives. Due to the lack of professional home office space, countless people have been forced to set up makeshift workspaces in living rooms, kitchens, and bedrooms. In addition, with schools being closed many teleworking parents are distracted as a result of having to supervise their children while working. It is likely that these changes have resulted in decreased productivity and increased stress in teleworkers.

Several studies [42–44] have shown the negative impact of teleworking on spare time, converting a place for rest into a workplace, which can be exhausting for teleworkers. In particular, significant negative effects, such as social isolation and stress levels, were observed from the second or third day of teleworking. Unlike teleworkers, those who insisted on working in the office were more distressed by economic pressures because they were more likely to be non-managers, informal staff, and single. To address these issues, current research in occupational

health psychology suggests that the adverse effects of teleworking can be mitigated through multifactorial measures such as adequate technical support, ongoing socially supportive communication within the team, personal health promotion interventions, and flexible working hours [45, 46].

Compared with the OO group, the OT and TT groups showed more physical symptoms after the emergency statement due to teleworking. This may be related to an unprofessional office environment. Alan et al. [47] suggested that the lack of space at home and inappropriate posture during work were common causes of complaints about sore eyes, fatigue, neck pain, and wrist pain. Telework inevitably leads to increased use and dependence on computers and other devices, which leads to computer vision syndrome, including internal ocular symptoms (strain and ache), external ocular symptoms (dryness, irritation, and burning), visual symptoms (blurriness and double vision), and musculoskeletal symptoms (neck and shoulder pain) [48]. Of these, musculoskeletal disorders (MSDs) require particular attention. MSD is the leading cause of disability worldwide, with a high prevalence throughout life, and has a tremendous impact on individuals and society [49]. Several studies [50, 51] have shown that inappropriate workstation conditions, especially chair height, and incorrect arm and back support, are important factors in the development of MSDs. Celik et al. [52] suggested that to protect workers from MSDs, it is important to make ergonomic adjustments to the work environment and take steps to ensure healthy lifestyle behaviors.

## Conclusion

Telework, as the mainstream of working in the future, has been widely discussed even before the pandemic. We have studied the health problems caused by teleworking under the specific circumstances of a global pandemic, when workers faced inadequate physical exercise, increasing alcohol consumption, tendency of depression and anxiety, aggravated work-family conflict, stress in confined spaces, and health concerns. With new epidemic peaks striking many countries around the world, the health problems associated with teleworking should be given adequate attention.

## Supporting information

**S1 File. Study questionnaire.**
(DOCX)

## Acknowledgments

We express our profound gratitude to our study participants and research members for their generous contributions to this study.

## Author Contributions

**Conceptualization:** Tomohisa Nagata, Naoto Fukutani, Masato Tezuka.

**Data curation:** Qian Niu.

**Formal analysis:** Qian Niu.

**Investigation:** Naoto Fukutani.

**Methodology:** Tomohisa Nagata, Masato Tezuka.

**Resources:** Tomohisa Nagata, Naoto Fukutani, Masato Tezuka.

**Supervision:** Tomoki Aoyama.

**Writing – original draft:** Qian Niu.

**Writing – review & editing:** Qian Niu, Tomohisa Nagata, Naoto Fukutani, Masato Tezuka, Kanako Shimoura, Momoko Nagai-Tanima, Tomoki Aoyama.

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
