## [Decision Letter · Decision Letter 0]

28 Jun 2021

PONE-D-21-17445

Health effects of immediate telework introduction during the COVID-19 era in Japan: A cross-sectional study

PLOS ONE

Dear Dr. Aoyama,

Thank you for submitting your manuscript to PLOS ONE. After careful consideration, we feel that it has merit but does not fully meet PLOS ONE’s publication criteria as it currently stands. Therefore, we invite you to submit a revised version of the manuscript that addresses the points raised during the review process.

We look forward to receiving your revised manuscript.

Kind regards,

Jianguo Wang, PhD

Academic Editor

PLOS ONE

Journal Requirements:

We note that one or more of the authors are employed by a commercial company:  BackTech Inc.

2.1. Please provide an amended Funding Statement declaring this commercial affiliation, as well as a statement regarding the Role of Funders in your study. If the funding organization did not play a role in the study design, data collection and analysis, decision to publish, or preparation of the manuscript and only provided financial support in the form of authors' salaries and/or research materials, please review your statements relating to the author contributions, and ensure you have specifically and accurately indicated the role(s) that these authors had in your study. You can update author roles in the Author Contributions section of the online submission form.

2.2. Please also provide an updated Competing Interests Statement declaring this commercial affiliation along with any other relevant declarations relating to employment, consultancy, patents, products in development, or marketed products, etc. 

<h1> </h1>

6. Please include your tables as part of your main manuscript and remove the individual files. Please note that supplementary tables should remain uploaded as separate "supporting information" files.

Reviewers' comments:

Reviewer's Responses to Questions

**Comments to the Author**

1. Is the manuscript technically sound, and do the data support the conclusions?

Reviewer #1: Yes

Reviewer #2: Yes

2. Has the statistical analysis been performed appropriately and rigorously? 

Reviewer #1: Yes

Reviewer #2: Yes

3. Have the authors made all data underlying the findings in their manuscript fully available?

Reviewer #1: Yes

Reviewer #2: Yes

4. Is the manuscript presented in an intelligible fashion and written in standard English?

Reviewer #1: Yes

Reviewer #2: Yes

5. Review Comments to the Author

Reviewer #1: The study by Niu et al. provides both current and interesting insights into potential health issues associated with teleworking during the COVID-19 pandemic. Furthermore, the study provides a distinction between those that newly switched to teleworking and those that are already used to this work modality with a focus on selected health outcomes.

Despite the good quality of the manuscript in its current form, I would like to encourage the authors to discuss the following aspects.

Reviewer Comment 1: For the present study, the authors collected survey data over a period of three months. People answering the survey at the beginning of the period might not be as familiar with the teleworking situation as those that answered it at the end of the period. People might have more struggles at the beginning regarding work-family conflicts, depression and anxiety, or physical symptoms compared to a later point, where they might have learnt to cope with the situation. May the long survey period have had an impact on the responses? Did the authors perform any sensitivity analyses investigating the impact of the time (or month) the survey was answered on some of the main outcomes?

Reviewer Comment 2: (Lines 312-324) The authors provide a constructive discussion about potential explanations for the increase in symptom scores for the OT and TT groups. While an unprofessional work environment seems a plausible and likely explanation for OT, it does not seem sufficient for the TT group. These participants should be used to teleworking and working in this exact environment. What could be additional explanations for their increase in symptom score? Might the fact that co-habitants are more frequently at home during the working hours, thereby inducing a new work environment or the decrease in exercise habits be potential explanations? Are there any correlations between symptom score and cohabitants/exercise habits for the TT group? Finally, the increase in work hours in over 50% of the participants might affect the symptom score as well.

Minor Comments:

Change line 25 to: […] emergency state […].

Line 55: Please provide a reference.

Line 208: Please use only one decimal place for the mean age as you did for SD.

Methods:

1) Physical symptoms: It is unclear from the methods section that physical symptoms were assessed retrospectively as well (“before” measurement point mentioned in table 5). How was “before” defined in the questionnaire?

2) Please describe all statistical tests used in the “Data processing and analysis” section (i.e., Kruskal-Wallis H test).

Discussion: I would encourage the authors to include a short paragraph including the main results of the study at the beginning of the discussion.

Line 264: “Our results are […].” The results regarding what? Providing one keyword in the first sentence would help the reader to directly understand what the paragraph is about.

Line 285: Please provide references to those previous studies.

Change line 319 to: MDS are the leading [...].

Conclusion: Since the current study also provided noteworthy information about statistical differences between OT and TT, I would recommend mentioning that being used to teleworking vs. being new to teleworking might have an impact on potential health consequences. (e.g., significant difference in mental health between OT and TT).

Table 5: While I understand the idea behind using the total score as a feasible summary measure of physical symptoms for the respective groups, this number is greatly dependent on the number of subjects in each group. Would using total score / n (e.g., for OO: 91/86=1.06 vs. 106/86=1.23) instead of the total score alone make the results better understandable? In my eyes, this could facilitate between-group comparisons for the reader.

Limitations: Finally, to complete this interesting manuscript, I would recommend to the authors including a short paragraph of this study’s limitations.

Reviewer #2: This study is to identify determine the health effects of teleworking during an emergency statement with COVID-19 in Japan. It is an interesting and important topic in the field of organization behavior. There are some comments as for improving the study.

(1) Introduction: This paper does not demonstrate the importance or value of this work or research. Please demonstrate the theoretical and practical meaning of this work.

(2) Literature review: Insufficient literature review about telework. This paper needs to clear the base of theory, and the between telework, and health factors.

(3) Please add the cited information about measure, such as telework, lifestyle change, and Physical symptoms.

(4) The implications need to improve further.

6. PLOS authors have the option to publish the peer review history of their article (what does this mean?). If published, this will include your full peer review and any attached files.

Reviewer #1: **Yes: **Fabian Schwendinger

Reviewer #2: No

---

## [Author Response · Author response to Decision Letter 0]

20 Jul 2021

Responses to the comments of Reviewer #1

● Reviewer Comment 1: For the present study, the authors collected survey data over a period of three months. People answering the survey at the beginning of the period might not be as familiar with the teleworking situation as those that answered it at the end of the period. People might have more struggles at the beginning regarding work-family conflicts, depression and anxiety, or physical symptoms compared to a later point, where they might have learnt to cope with the situation. May the long survey period have had an impact on the responses? Did the authors perform any sensitivity analyses investigating the impact of the time (or month) the survey was answered on some of the main outcomes?

Response: Thanks for your nice suggestion. Actually, this study is a retrospective cross-sectional survey study in which we administered questionnaires to 5,000 workers between June 7 and September 7, 2020, asking them to recall their health conditions related to telework during the emergency state from April 7 to May 25, 2020. On the other hand, current survey especially focused on the beginning of the working style shift to telework. So we use this specific short span and we didn't perform any sensitivity analysis.

We are so sorry that the narrative was ambiguous and I have revised that part as follows and highlighted it in the manuscript.

○ We administered questionnaires to 5000 workers between June 7 and September 7, 2020, asking them to recall their health conditions related to telework during the emergency state from April 7 to May 25, 2020. 

● Reviewer Comment 2: (Lines 312-324) The authors provide a constructive discussion about potential explanations for the increase in symptom scores for the OT and TT groups. While an unprofessional work environment seems a plausible and likely explanation for OT, it does not seem sufficient for the TT group. These participants should be used to teleworking and working in this exact environment. What could be additional explanations for their increase in symptom score? Might the fact that co-habitants are more frequently at home during the working hours, thereby inducing a new work environment or the decrease in exercise habits be potential explanations? Are there any correlations between symptom score and cohabitants/exercise habits for the TT group? Finally, the increase in work hours in over 50% of the participants might affect the symptom score as well.

Response: Thanks for your comment. It is indeed unreasonable for individuals who are already familiar with teleworking to still experience a similar degree of physical symptoms as new teleworkers, attributed only to an unprofessional work environment. We agree with your suggestion that it may be related to cohabitants and lack of exercise. But there are weak correlations between symptom score and cohabitants/exercise habits/work hours used by Kendall's tau-b correlation coefficient, the value is 0.055, -0.052, 0.062 respectively. Therefore, we believe that there is no single cause of physical symptoms in the TT group, as shown in Tables 3 and 4, which show statistically significant differences in both work-family conflict and mental health in the TT group compared to the OT group. Those who were already familiar with telework seemed to demonstrate more health issues compared to new teleworkers, which deserves our attention. 

We added a paragraph at the end of “Discussion” as follows and highlighted the text in the manuscript. 

○ “However, the TT group, which should have been accustomed to teleworking mode and equipped with a relatively professional working environment before the emergency state compared to the OT group, still showed more physical symptoms. It may be related to the fact that the cohabitants also worked remotely and thus induced a new uncomfortable environment. Reduced exercise habits due to working from home during the epidemic and over half of them reporting longer working hours may also have contributed. But there are weak correlations between symptom score and cohabitants/exercise habits/work hours used by Kendall's tau-b correlation coefficient. Therefore, we believe that physical symptoms in the TT group are not caused by a single reason, as shown in Tables 3 and 4, which show statistically significant differences in both work-family conflict and mental health in the TT group compared to the OT group. Those who were already familiar with telework seemed to demonstrate more health issues compared to new teleworkers, which deserves our attention. ”

● Minor Comments:

Change line 25 to: […] emergency state […].

Line 55: Please provide a reference.

Line 208: Please use only one decimal place for the mean age as you did for SD.

Response: Thanks for your comments. According to your comments, We have revised these sentences in red color.

● Methods:

● 1) Physical symptoms: It is unclear from the methods section that physical symptoms were assessed retrospectively as well (“before” measurement point mentioned in table 5). How was “before” defined in the questionnaire?

Response: This study is a retrospective cross-sectional study, the participants were asked to recall the health conditions before and after the emergency state. And the emergency state is from April 7 to May 25, 2020, data collection is June 7 and September 7, 2020. 

We put the information in this paragraph:

○ Participants retrospectively answered separately whether they experienced the corresponding physical symptoms before and after the emergency state, and cumulative scores were used for before and after comparisons.

● 2) Please describe all statistical tests used in the “Data processing and analysis” section (i.e., Kruskal-Wallis H test).

Response: Thanks for your comment. We have described all statistical tests with Kruskal-Wallis H test.

We put the information on this paragraph: 

○ The Kruskal-Wallis H test was used to compare the ordered categorical variables before and after the emergency state.

● Discussion: I would encourage the authors to include a short paragraph including the main results of the study at the beginning of the discussion.

Response: I appreciate your good suggestion to add a short paragraph including the main result to make readers understand easier. Thank you very much.

We added a paragraph at the beginning of “Discussion” as follows and highlighted the text in the manuscript. 

○ According to the results, we found that the workers' residence, marital status, management position, and employee status affected their choice of work method. Teleworkers' habits changed more during the emergency than those of office workers. Of the individuals belonging to the office-telework (OT) group, 67.0% exercised less. In the OT group, teleworkers were less satisfied with their telework than those in the telework-telework group (TT group), and their working hours and meeting hours both increased. In contrast to the two other groups, the TT group had a greater work-family conflict. The majority of individuals reported experiencing stress in the past 30 days, but the degree of anxiety and depressive symptoms varied. Additionally, all teleworkers experienced adverse physical symptoms during and immediately after the emergency.

● Line 264: “Our results are […].” The results regarding what? Providing one keyword in the first sentence would help the reader to directly understand what the paragraph is about.

Response: Thanks for your comment. We revised this sentence as follows and highlighted it in the manuscript.

○ Our results showed that more than half of the teleworkers had reduced exercise habits during the emergency, inconsistent with the predictive model of Brand et al.

● Line 285: Please provide references to those previous studies.

Response: We added the references to those previous studies in red color.

● Change line 319 to: MDS are the leading [...].

Response: We changed this sentence and thank you very much.

● Conclusion: Since the current study also provided noteworthy information about statistical differences between OT and TT, I would recommend mentioning that being used to teleworking vs. being new to teleworking might have an impact on potential health consequences. (e.g., significant difference in mental health between OT and TT).

Response: Thanks for your suggestion. That’s right it's better to add the different information between OT and TT. We revised the conclusion as follows and highlighted it in the manuscript.

○ Telework, as the mainstream of working in the future, has been widely discussed even before the pandemic. We have studied the health problems caused by teleworking under the specific circumstances of a global pandemic, when workers faced inadequate physical exercise, increasing alcohol consumption, tendency of depression and anxiety, aggravated work-family conflict, stress in confined spaces, and health concerns. Moreover, new teleworkers reported more work-family conflicts and mental problems than those who were already used to telework. With new epidemic peaks striking many countries around the world, the health problems associated with teleworking should be given adequate attention.

● Table 5: While I understand the idea behind using the total score as a feasible summary measure of physical symptoms for the respective groups, this number is greatly dependent on the number of subjects in each group. Would using total score / n (e.g., for OO: 91/86=1.06 vs. 106/86=1.23) instead of the total score alone make the results better understandable? In my eyes, this could facilitate between-group comparisons for the reader.

Response: Thank you very much. According to your suggestion, we revised the demonstration of this value as highlighted in the manuscript.

● Limitations: Finally, to complete this interesting manuscript, I would recommend to the authors including a short paragraph of this study’s limitations.

Response: Thanks for your comment. We added a paragraph as follows and highlighted it in the manuscript.

○ Limitation: This study was a retrospective cross-sectional study, which inevitably produces recall bias. Because it was an emergency state for more than a month, the participants were undergoing dynamic changes from initial entry into telework to familiarization with the telework environment, but our study lacked follow-up observations for this month.

Reviewer #2: 

● (1) Introduction: This paper does not demonstrate the importance or value of this work or research. Please demonstrate the theoretical and practical meaning of this work.

Response: Thanks for your comments. To emphasize this paper’s importance and value, we revised the paragraph at the end of introduction as follows and highlighted it in the manuscript.

○ While telework is beneficial when dealing with outbreaks, we believe that it may have adverse effects on both individual and public health if not used correctly. The work-family conflict, psychological stress, and physical harm caused by teleworking cannot be ignored. However, the ability to make sound policy recommendations regarding telework depends on extensive and qualitative research. Surprisingly, little research has been conducted on the association between telework and health-related performance, especially during the pandemic. Other than that, we believe that the impact of COVID-19 on changes in work behavior will be profound, and that as people become accustomed to teleworking during the epidemic, it will become the dominant mode of work after the epidemic. This study aimed to determine the health problems associated with teleworking during a pandemic, thereby improving the well-being of teleworkers and providing evidence to support subsequent related studies. 

● (2) Literature review: Insufficient literature review about telework. This paper needs to clear the base of theory, and the between telework, and health factors.

Response: Thanks for your comment. We conducted a literature review using telework as an entry point to introduce health issues as one of its side effects. In this particular period, during which Japan was in emergency state for the first time, the health issues of new teleworkers as a group deserve our more attention. We stated this in the last paragraph of the introduction.

● (3) Please add the cited information about measure, such as telework, lifestyle change, and Physical symptoms.

Response: Thanks for your comment. We have added the cited information about measures, such as work-family conflict and depression and anxiety. Other than that lifestyle changes, telework and physical symptoms are investigated by the questionnaire we set up for the emergency state and did not use any relevant scales.

● (4) The implications need to improve further.

Response: Thank you very much. To emphasize the implications, we revised the conclusion as follows and highlighted it in the manuscript.

○ Telework, as the mainstream of working in the future, has been widely discussed even before the pandemic. We have studied the health problems caused by teleworking under the specific circumstances of a global pandemic, when workers faced inadequate physical exercise, increasing alcohol consumption, tendency of depression and anxiety, aggravated work-family conflict, stress in confined spaces, and health concerns. Moreover, those already used to telework reported more work-family conflicts and mental problems than new teleworkers. With new epidemic peaks striking many countries around the world, although telework has become an effective alternative to traditional office work, the health problems associated with teleworking should be given adequate attention. Work is an integrated part of our social connection, and the isolation and physical impairment associated with telework bring new challenges for individuals and society alike. Our study examines the health issues arising from telework during the Japanese emergency state, and we will follow up on it in the post-epidemic era.

---

## [Decision Letter · Decision Letter 1]

9 Aug 2021

Health effects of immediate telework introduction during the COVID-19 era in Japan: A cross-sectional study

PONE-D-21-17445R1

Dear Dr. Aoyama,

We’re pleased to inform you that your manuscript has been judged scientifically suitable for publication and will be formally accepted for publication once it meets all outstanding technical requirements.

Kind regards,

Jianguo Wang, PhD

Academic Editor

PLOS ONE

Additional Editor Comments (optional):

Reviewers' comments:

Reviewer's Responses to Questions

**Comments to the Author**

1. If the authors have adequately addressed your comments raised in a previous round of review and you feel that this manuscript is now acceptable for publication, you may indicate that here to bypass the “Comments to the Author” section, enter your conflict of interest statement in the “Confidential to Editor” section, and submit your "Accept" recommendation.

Reviewer #1: All comments have been addressed

Reviewer #2: All comments have been addressed

2. Is the manuscript technically sound, and do the data support the conclusions?

Reviewer #1: Yes

Reviewer #2: Yes

3. Has the statistical analysis been performed appropriately and rigorously? 

Reviewer #1: Yes

Reviewer #2: Yes

4. Have the authors made all data underlying the findings in their manuscript fully available?

Reviewer #1: Yes

Reviewer #2: Yes

5. Is the manuscript presented in an intelligible fashion and written in standard English?

Reviewer #1: Yes

Reviewer #2: Yes

6. Review Comments to the Author

Reviewer #1: (No Response)

Reviewer #2: Thank you for your revision and improvement of the article. This study was explored to determine the health effects towards teleworking during the COVID-19 ear in Japan.

7. PLOS authors have the option to publish the peer review history of their article (what does this mean?). If published, this will include your full peer review and any attached files.

Reviewer #1: **Yes: **Fabian Schwendinger

Reviewer #2: No

---

## [Editor Report · Acceptance letter]

29 Sep 2021

PONE-D-21-17445R1 

Health effects of immediate telework introduction during the COVID-19 era in Japan: A cross-sectional study 

Dear Dr. Aoyama:

I'm pleased to inform you that your manuscript has been deemed suitable for publication in PLOS ONE. Congratulations! Your manuscript is now with our production department. 

Kind regards, 

on behalf of

Dr. Jianguo Wang 

Academic Editor

PLOS ONE